# Persona-aware Generative Model for Code-mixed Language

## Abstract

Code-mixing and script-mixing are prevalent across online social networks and multilingual societies. However, a user's preference toward code-mixing depends on the socioeconomic status, demographics of the user, and the local context, which existing generative models tend to ignore while generating code-mixed texts. In this work, we make a pioneering attempt to develop a persona-aware generative model to generate texts resembling real-life code-mixed texts of individuals. We propose `PARADOX`, a persona-aware generative model for code-mixed text generation, which is a novel Transformer-based encoder-decoder model that encodes an utterance conditioned on a user's persona and generates code-mixed texts *without* monolingual reference data. We propose an alignment module that re-calibrates the generated sequence to resemble real-life code-mixed texts. `PARADOX` generates code-mixed texts that are semantically more meaningful and linguistically more valid. To evaluate the personification capabilities of `PARADOX`, we propose four new metrics – CM BLEU, CM Rouge-1, CM Rouge-L and CM KS. On average, `PARADOX` achieves 1.6 points better CM BLEU, 47% better perplexity and 32% better semantic coherence than the non-persona-based counterparts.

## 1 Introduction

Code-mixing (*aka* code-switching) appears when two or more languages are used interchangeably in a single utterance. It is common in multilingual societies like India, where more than 24% of the population speaks in more than one language (Sengupta et al., 2021). Code-mixing is even more prevalent on social media. Informal usage of code-mixed languages on social media platforms like Twitter, Facebook, YouTube, and other online social networks gives rise to *script-mixing*, in which a user can use a single script (*e.g.,* Roman) or multiple scripts (*e.g.,* Devanagari for Hindi and Roman for English) within the same text (Srivastava et al., 2020). Recent literature has made significant efforts to understand syntactic structure and semantics from code-mixed texts (Singh et al., 2018a;b; Sengupta et al., 2022b). Similar attempts have been made for pragmatic tasks – humour, sarcasm and hate detection in the code-mixed regime (Sengupta et al., 2022a; Bansal et al., 2020).

Text generation models need to understand the syntax and semantics of texts and preserve semantic coherence during generation. Previous studies utilized recurrent neural networks with generative models (Zhang et al., 2017), as well as self-attention-based pre-trained language models (Zhang et al., 2020) for generating monolingual texts. However, such an effort is limited in case of code-mixing. Previously, linguistic theories (Pratapa et al., 2018; Gupta et al., 2020), transfer learning (Gupta et al., 2020), and autoencoding (Samanta et al., 2019) based approaches have been used to generate code-mixed texts from parallel corpora or reference data. However, none of these methods incorporates user information while generating code-mixed texts. Unlike traditional languages, code-mixing is a derived language whose adoption depends on different socioeconomic, demographic, and linguistic factors (Rudra et al., 2016; Parshad et al., 2016). Figure 1 demonstrates the code-mixing behaviour among Indian users on Twitter and YouTube in terms of adoption and patterns in code-mixing. We visualize the mean and standard deviation of the Code-Mixing Index (CMI) (Das & Gambäck, 2014) and the length of tweets/comments posted by different users. The distributions show how different users conceive and prefer code-mixing.

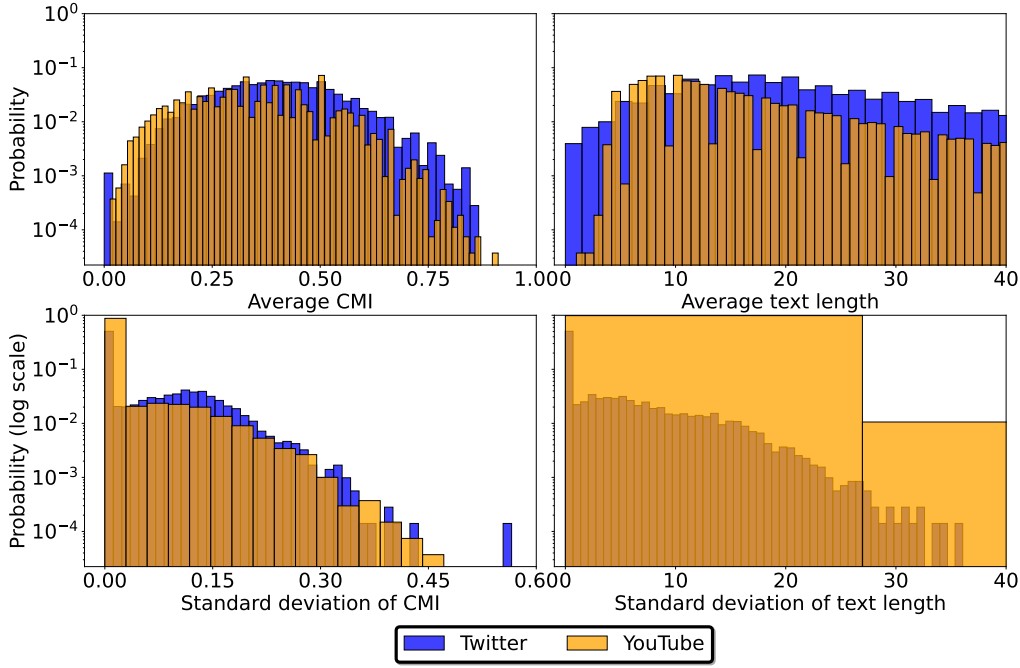

Figure 1: User-specific distribution of Code-Mixing Index (CMI) and text lengths across different platforms. CMI is calculated as the fraction of minority language words in a text. For instance, the CMI of the text "I don't want your nautanki" ("I don't want your gimmick") is $\frac{1}{5} = 0.2$, the fraction of Hindi (minority language in this example) words in the text. Texts skewed toward monolingualism, *i.e.,* having an unequal proportion of words between different languages, tend to have lower CMI than multilingual texts.

User persona plays a vital role in generation models, particularly in personalized generation settings, such as conversational agents and recommendation engines. Several studies have contributed towards persona-based dialogue generation (Zheng et al., 2019; Wang et al., 2021), personalized story generation (Chandu et al., 2019), and other sub-tasks in text generation. Being a conversational language, personification of code-mixing could be deemed appropriate for conversational systems such as recommender engines, mental health counselling bots, and event booking applications.

This motivates us to develop `PARADOX`, a novel **p**ersona-**a**ware gene**ra**tive mo**d**el for c**o**de-mi**x**ed text generation. It aims to generate personalized code-mixed texts by leveraging users' historical utterances. It uses a Transformer-based encoder-decoder architecture to learn the semantics of code-mixed generation. The model utilizes a novel persona encoder to encode a user persona from their behavioural preferences. Instead of projecting the user's persona onto a static space, `PARADOX` projects it onto a probabilistic latent space and captures the contextual persona based on their historical persona. Additionally, `PARADOX` uses an alignment module to re-align decoder outputs to generate coherent texts.

We evaluate `PARADOX` against the vanilla Transformer in terms of both the quality and coherence of generated texts. To quantify the extent of personification in the code-mixed generation, we propose four metrics – CM BLEU, CM Rouge-1, CM Rouge-L, and CM KS (here CM stands for code-mixing). On average, `PARADOX` achieves 1.6 points better CM BLEU than the non-persona counterpart. We also conduct a detailed human evaluation, concluding that `PARADOX`-generated code-mixed texts are 32% more semantically coherent than that of the vanilla Transformer model. `PARADOX` can imitate a user's linguistic preference 4% better than the non-persona-based Transformer model. Our empirical analyses also highlight the effectiveness of `PARADOX` over pre-trained large language models. On average, `PARADOX` achieves 4.2 points better CM BLEU, 11 points better CM Rouge-1, and 9.6 points better CM Rouge-L than the pre-trained Llama 2 model (Touvron et al., 2023).

**Contributions.** The major contributions of this paper are summarized below:

- We make a pioneering effort in utilizing user persona in code-mixed text generation. Compared to existing approaches, `PARADOX` does not require parallel corpora or reference data for code-mixed text generation.
- We propose a probabilistic persona encoder module that learns the latent persona of a user from historical contexts. `PARADOX` captures the user persona implicitly and does not require explicit persona features such as user demographic information to encode user behaviours.
- We design an alignment module to automatically induce alignments between different subwords. Empirical results show that it improves the coherence of generated texts.
- We propose three metrics influenced by supervised machine translation – CM BLEU, CM Rouge-1, and CM Rouge-L for evaluating the personification of code-mixed generation models. We also propose CM KS as a distance measure to evaluate code-mixing generation models.
- We further explore knowledge distillation and meta-knowledge distillation in the context of text generation and present the effectiveness of knowledge transfer from language models pre-trained on multilingual texts.
- Finally, we collect a large-scale longitudinal dataset from Twitter and YouTube, primarily monolingual Hindi and Hindi-English code-mixed texts. The datasets will be valuable for code-mixing research.

**Reproducibility.** The supplementary material comprises the source code and datasets.

## 2 Related Works

**Rule-based and Linguistic Approaches.** Code-mixed text generation has garnered much interest in recent times. Pratapa et al. (2018) explored *equivalence constraint* (EC) theory to generate Spanish-English code-mixed texts from monolingual corpora. Their linguistic theory-based approach showed superiority over recurrent neural networks in complex code-mixed text generation. Rizvi et al. (2021) developed GCM, a toolkit that utilizes different linguistic theories to generate code-mixed texts. Motivated by embedding matrix theory, Srivastava & Singh (2021) proposed rule-based methods to generate Hindi-English code-mixed texts. Santy et al. (2021) utilized parse tree structures within the monolingual texts for generating code-mixed texts. Alternative approaches use generative models – generative adversarial networks (Goodfellow et al., 2020), or variational autoencoder (VAE) (Kingma & Welling, 2014). Towards this, Garg et al. (2018) explored recurrent neural networks with SeqGAN pre-training for generating Mandarin-English code-mixed data. Samanta et al. (2019) developed a VAE-based method to generate realistic and coherent Hindi-English code-mixed texts. Other classes of code-mixed text generation models explore alignment within parallel corpora for code-mixed generation. Notably, Winata et al. (2019); Tan & Joty (2021) explored word alignments and candidate selection from parallel corpora for generating synthetic code-mixed texts. Amin et al. (2023) explored word alignments for generating Marathi-English code-mixed text generation. On a similar attempt, Dowlagar & Mamidi (2021) explored gated convolutional encoder-decoder models to identify the compositional structure and translate English texts to Hinglish.

**Pre-trained Models.** With the inception of self-attention (Vaswani et al., 2017), several attempts have been made to develop large pre-trained models showing exceptional performances in semantic and generative tasks. Among these methods, multilingual models, such as XLM (Conneau & Lample, 2019), XLM-RoBERTa (Conneau et al., 2020), and mBART (Liu et al., 2020) have shown noticeable performance even on low-resource languages. Recently, MuRIL (Khanuja et al., 2021) was proposed, superseding the performances of multilingual-BERT (Devlin et al., 2019) on different syntactic and semantic tasks on a diverse set of low-resource languages. Gupta et al. (2020) devised a semi-supervised approach to transfer knowledge from XLM to generate synthetic Hindi-English code-mixed texts. Gautam et al. (2021) explored a pre-trained mBART model for generating Hindi-English code-mixed texts. Jawahar et al. (2021) explored multilingual text-to-text models with curriculum learning for generating Hindi-English code-mixed texts. They pre-trained an encoder-decoder model on synthetic code-mixed texts, which improved the generation quality on the gold code-mixed dataset. In a recent study, Yong et al. (2023) explored multilingual large language models (LLMs) for generating code-mixed texts in a zero-shot setting. They explored InstructGPT, ChatGPT (Ouyang et al., 2022), BLOOMZ (Muennighoff et al., 2022) and Flan-T5-XXL (Chung et al., 2022) for generating code-mixed texts in Indonesian, Malay, Chinese, Tagalog, Vietnamese, Tamil, and Singlish. They further emphasized the importance of better prompt templates and language pairing for generating more coherent and natural code-mixed texts.

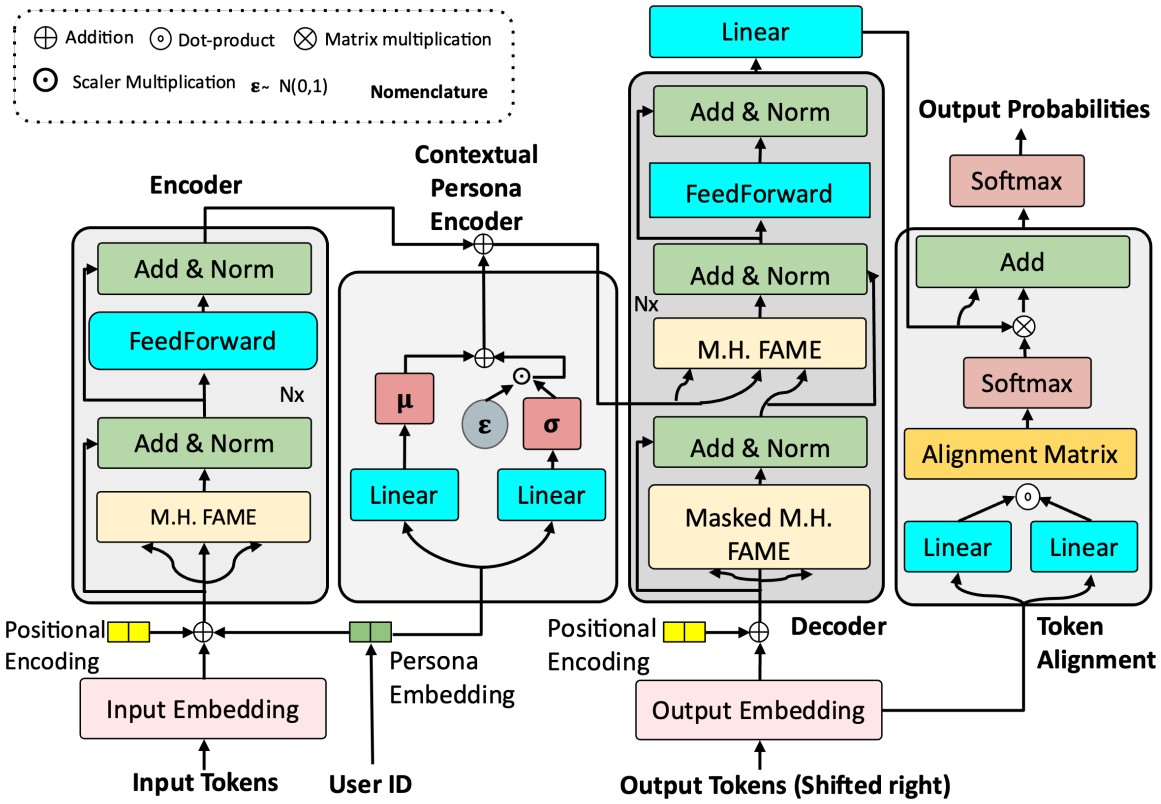

Figure 2: `PARADOX`: Transformer encoder-decoder architecture with persona encoder (multi-headed (M.H.) fused attention (FAME)).

**Major Limitations of Existing Studies.** Despite their popularity in several generative applications, personalization remains neglected in the code-mixed generation. The existing code-mixed generation models utilize parallel corpora to understand the switching patterns and generate code-mixed texts synthetically. These limitations motivate us to develop a code-mixed language generation model that can automatically learn the language's semantics and capture the linguistic preferences of users while generating texts. Our proposed method, `PARADOX`, improves the quality of generated texts and preserves the real-life phenomenon of code-mixing among users.

## 3 Proposed Methodology

Here, we explain `PARADOX`, and the code-mixing (CM, henceforth) generation process utilizing user persona. `PARADOX` utilizes a persona encoder to implicitly encode a user's persona based on her historical utterances. It further projects the persona onto a probabilistic latent space to capture the user's contextual persona. `PARADOX` employs an alignment module to re-calibrate the output sequences that help our model understand the language of code-mixing and enable the model to generate coherent texts.

### 3.1 `PARADOX` Architecture

`PARADOX`, as shown in Figure 2, consists of four components – (a) an encoder, (b) a contextual persona encoder, (c) a decoder, and (d) an alignment module.

### 3.1.1 The Encoder with FAME

The encoder is used to jointly learn the semantics of a code-mixed text and a user's global persona based on the previous comments/tweets. A comment/tweet $\mathbf{X}_u$ by user $u$ is first tokenized using byte-pair encodings (Sennrich et al., 2016) into $\langle x_1, x_2, ..., x_n \rangle$. The initial contextual embedding of a token $x_i$ is conditioned with the user persona as

$$\widetilde{Emb_{(x_i,u)}} = Emb_{x_i} + PE_i + Emb_u \tag{1}$$

where $Emb_{x_i}$ is the initial token embedding, $PE_i$ is the positional encoding at position $i$, and $Emb_u$ is the user's global persona embedding captured through a static embedding layer. We use a stacked encoder, in which each encoder consists of multi-headed fused attention (FAME), followed by a residual, layer-normalization, and pointwise feed-forward layers. Sengupta et al. (2021) introduced FAME by combining scaled dot-product attention (Vaswani et al., 2017) and outer-product attention (Le et al., 2020) and showed to be effective in capturing both semantics and morphology of code-mixed texts.

### 3.1.2 Contextual Persona Encoder

We project the static persona embedding onto a probabilistic latent space for generating the contextual persona embedding for each user in a given context. We hypothesize that each user has a static (global) persona and a contextual (local) persona. The motivation behind projecting the persona embedding to a latent space is to capture the contextual perturbations in the user persona. For example, a user can have a generally positive outlook (global) on a particular concept; however, given a specific situation, the extent of the positive outlook (local) might change. Formally, we generate a contextual persona embedding

$$\widetilde{Emb_u} \sim q_\phi(z|Emb_u) = \mathcal{N}(\mu_u, \sigma_u^2). \tag{2}$$

Towards this, we define two linear projection matrices to learn the distribution location and scale parameters as

$$\mu_u = Emb_u.W_\mu \text{ and } \sigma_u = Emb_u.W_\sigma.$$

Following the reparameterization trick (Kingma & Welling, 2014), we define the final generated persona encoding as:

$$\widetilde{Emb_u} = \mu_u + \epsilon_u \odot \sigma_u \tag{3}$$

where $\epsilon_u$ is the random noise, independently drawn from $\mathcal{N}(0,1)$. We obtain the hidden representation of token $x_i$ conditioned on the contextual persona encoding as

$$\widetilde{h_{(x_i,u)}} = h_{(x_i,u)} + \widetilde{Emb_u} \tag{4}$$

where $h_{(x_i,u)}$ is the final hidden representation obtained from the final layer of the encoder.

### 3.1.3 The Decoder

We adopt the Transformer decoder conditioned on the contextual user persona. Similar to the original Transformer decoder, we use a stacked decoder initialized with the encoded output sequence. Drawing the motivation from autoregressive generative language models like GPT2 (Radford et al., 2019), the decoder's objective is to predict the next token, conditioned on all the previous tokens. The input to the decoder is the encoded input sequence added with positional encoding. Each decoder block consists of masked multi-headed FAME, a residual connection, and a normalization layer. We also deploy multi-headed FAME to attend to each decoder token with the encoded input tokens $\widetilde{h_{(x_i,u)}}$. For each decoder input position $j$, we generate a hidden representation $h_{(j,u)}^{(dec)} \in \mathbb{R}^{|V|}$, representing the output token at $(j+1)^{th}$ position; $|V|$ is the vocabulary size of the decoder.

### 3.1.4 The Alignment Module

The final layer of PARADOX is an alignment module that learns the latent alignment matrix and re-aligns the outputs generated by the decoder. The primary objective behind using alignments in generative models is explicitly learning the global semantic similarity between different tokens. We use two projection matrices, $W^Q$ and $W^K$, to project the decoder token embedding matrix $Emb^{(dec)}$ into two different subspaces. The alignment matrix is defined as,

$$\mathcal{A} = softmax\left(\frac{Q \cdot K^T}{\sqrt{d}}\right) \tag{5}$$

where $Q = Emb^{(dec)} \cdot W^Q$, $K = Emb^{(dec)} \cdot W^K$, and $d$ is the hidden size of the decoder. This operation resembles the scaled dot-product attention mechanism (Vaswani et al., 2017). However, as opposed to attention, we compute the global context by considering the original embedding space of all tokens. Finally, the re-aligned hidden representation is derived as,

$$\widetilde{h_{(j,u)}}^{(dec)} = h_{(j,u)}^{(dec)} \cdot \mathcal{A} + h_{(j,u)}^{(dec)} \tag{6}$$

This hidden representation is finally fed to a softmax layer to convert the outputs into probabilities.

For the sake of simplicity, we denote the combination of text encoder and persona encoder as 'encoder' and the combination of Transformer decoder and the alignment module as 'decoder' throughout this paper. The generative model is trained w.r.t. the decoder reconstruction cross-entropy loss. The contextual persona encoder gives rise to a variational KL-divergence loss. We use a variational hyperparameter $\lambda$ to assign its weightage in the final computed loss.

### 3.2 Training Curricula

To learn the model parameters, we primarily minimize the reconstruction loss on output sequence $\langle y_1, y_2, ..., y_m \rangle$ between defined as,

$$\mathcal{L}_1^{(x,u)} = \sum_{j=1}^{m} y_{(j,u)} \log(P_{\theta_{dec}}(y_{(j,u)}|\mathbf{Y}_{(0:j-1,u)}, \mathbf{X}_{1:n}, u)) \tag{7}$$

The output sequence is initialized with $y_0 =$ [CLS] token. The contextual persona encoder module arises a Kullback–Leibler divergence loss between the variational distribution and true posterior distribution, which can be derived (Kingma & Welling, 2014) to

$$\mathcal{L}_2^{(u)} = -\frac{1}{2}\sum_{k=1}^{d}\left(1 + 2 \cdot \log(\sigma_u^k) - (\mu_u^k)^2 - (\sigma_u^k)^2\right) \tag{8}$$

During training, we minimize the task-specific loss

$$\mathcal{L}^{(x,u)} = \mathcal{L}_1^{(x,u)} + \lambda \cdot \mathcal{L}_2^{(u)} \tag{9}$$

for each text and user id pair $(x, u) \sim \mathcal{D}$ on training data. The persona encoding weight $\lambda$ is a hyperparameter we set before running the experiments.

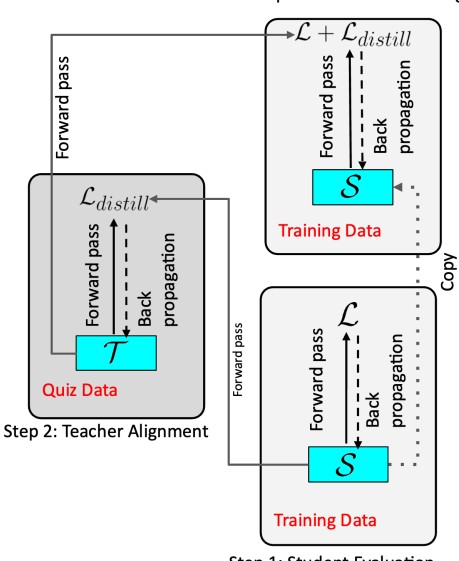

Figure 3: Meta knowledge distillation training curriculum for PARADOX.

In addition to the vanilla training, we define two other training curricula for jointly learning the code-mixed semantics and improving the reconstruction ability of our model by transferring knowledge from other large language models pre-trained on multilingual corpora.

---

**Algorithm 1:** Training with Distillation

---

**Require:** *student $\mathcal{S}$, teacher $\mathcal{T}$, training data $\mathcal{D}$*
**Require:** $\lambda$*, scaling factor; $\mu$, learning rate*
**while** *not converged* **do**

    Sample training batch $x, u \sim \mathcal{D}$;
    Encode $h_x^{\mathcal{T}}$ from teacher encoder;
    Encode $h_{(x,u)}^{\mathcal{S}}$ from student encoder;
    Calculate $\mathcal{L}_{distill}{}^{(x,u)}$ using Eqn. 10;
    Calculate $\mathcal{L}^{(x,u)} = \mathcal{L}^{(x,u)} + \lambda \cdot \mathcal{L}_{distill}{}^{(x,u)}$ using Eqn. 9;
    Update $\theta_{\mathcal{S}}$ with $\theta_{\mathcal{S}} \leftarrow \theta_{\mathcal{S}} - \mu \cdot \nabla_{\theta_{\mathcal{S}}} \mathcal{L}^{(x,u)}$;

**end**

---

### 3.2.1 Knowledge Distillation

Distillation (Hinton et al., 2015) (Distill, henceforth) is a widely popular method to transfer knowledge from large teacher models to smaller student models. In this study, we adopt large pre-trained language models as teacher models to transfer knowledge to a lighter student encoder of our model. Distillation not only improves the student model's learning ability but also improves our generative model's convergence. We define the distillation loss between teacher model $\mathcal{T}$ and the student model $\mathcal{S}$ as the average mean squared error (MSE) between the encoded representation of the input text as follows,

$$\mathcal{L}_{distill}{}^{(x,u)} = \frac{1}{n} \sum_{i=1}^{n} \left( \left\| h_{x_i}^{\mathcal{T}} - h_{(x_i,u)}^{\mathcal{S}} \right\|^2 \right) \tag{10}$$

We highlight the distillation process in Algorithm 1.

### 3.2.2 Meta Knowledge Distillation

Meta Knowledge Distillation (MetaDistill) (Zhou et al., 2022) is recently proposed to reduce the teacher-student gap during distillation. This technique improves student learning in a context where the teacher is trained out-of-domain. Most existing pre-trained language models are trained on large monolingual or multilingual corpora and lack knowledge of script-mixing. Under this context, MetaDistill can be effective for the adaptive learning of the teacher and student models. However, our adaptation of MetaDistill is slightly modified from the original technique, as we use a different learning objective for the teacher and student. The code-mixed MetaDistill process follows **student evaluation → teacher alignment → student relearning** (shown in Figure 3). In student evaluation, the student model is trained on the generative task without any teacher's supervision. In the next step, we align the teacher model to minimize distillation loss between the teacher and the student encoders on separate quiz data, keeping the student model weights frozen. Teacher alignment is necessary for the adaptive learning of the teacher in an out-of-domain code-mixed setting. Finally, the student is retrained with the teacher's supervision. In this step, we fine-tune the student model on the training data with task-specific (Equation 9) and distillation loss (Equation 10). We highlight our training curriculum with MetaDistil in Algorithm 2.

## 3.3 Code-Mixed Generation

We adopt an auto-regressive generation technique to generate new code-mixed texts for different users. To encode the user's historical persona, we use the user's last utterance (comment/tweet) in the encoder. Based on the decoder input, we devise a cold-start generation strategy. In the cold-start generation, we pass a seed word as input to the decoder and generate the rest of the text. We formally report the text generation process in Algorithm 3.

---

**Algorithm 2:** Training with Meta Distillation

---
**Require:** *Student $\mathcal{S}$, Teacher $\mathcal{T}$, training data $\mathcal{D}$, quiz data $\mathcal{Q}$*
**Require:** *Scaling factor: $\lambda$; learning rates: $\mu_1$, $\mu_2$, $\mu_3$*
**while** *not converged* **do**
  Sample training batch $x, u \sim \mathcal{D}$;
  Calculate $\mathcal{L}^{(x,u)}$ from Eqn. 9;
  Update $\theta_{\mathcal{S}}$ with $\theta_{\mathcal{S}} \leftarrow \theta_{\mathcal{S}} - \mu_1 \cdot \nabla_{\theta_{\mathcal{S}}} \mathcal{L}^{(x,u)}$, keeping $\theta_{\mathcal{T}}$ fixed;
  Sample quiz batch $\acute{x}, \acute{u} \sim \mathcal{Q}$;
  Encode $h_{\acute{x}}^{\mathcal{T}}$ from teacher encoder;
  Encode $h_{(\acute{x}, \acute{u})}^{\mathcal{S}}$ from student encoder;
  Calculate $\mathcal{L}_{distill}^{(\acute{x}, \acute{u})}$ using Eqn. 10;
  Update $\theta_{\mathcal{T}}$ with $\theta_{\mathcal{T}} \leftarrow \theta_{\mathcal{T}} - \mu_2 \cdot \nabla_{\theta_{\mathcal{T}}} \mathcal{L}_{distill}^{(\acute{x}, \acute{u})}$, keeping $\theta_{\mathcal{S}}$ fixed;
  Sample training batch $x, u \sim \mathcal{D}$;
  Encode $h_x^{\mathcal{T}}$ from teacher encoder;
  Encode $h_{(x,u)}^{\mathcal{S}}$ from student encoder;
  Calculate $\mathcal{L}_{distill}^{(x,u)}$ using Eqn. 10;
  Calculate $\mathcal{L}^{(x,u)} = \mathcal{L}^{(x,u)} + \lambda \cdot \mathcal{L}_{distill}^{(x,u)}$ using Eqn. 9;
  Update $\theta_{\mathcal{S}}$ with $\theta_{\mathcal{S}} \leftarrow \theta_{\mathcal{S}} - \mu_3 \cdot \nabla_{\theta_{\mathcal{S}}} \mathcal{L}^{(x,u)}$, keeping $\theta_{\mathcal{T}}$ fixed
**end**

---

---

**Algorithm 3:** Code-Mixed Text Generation with `PARADOX`

---
**Require:** *Trained model $\mathcal{M} = (enc, dec)$, user id $u$, historical utterance $x_u$, prompt word $\{w_1\}$, decoder vocabulary $V$*
**Require:** $max\_length \in \mathbb{N}$
$L \leftarrow \{w_1\}$;
$\tilde{w} \leftarrow \emptyset$;
$i = m$;
**while** $\tilde{w} \neq [SEP]$ *and* $i < max\_length$ **do**
  $h_{(x_u, u)} = enc(x_u, u)$;
  $P_{i+1} = dec(L, h_{(x_u, u)})$;
  $\tilde{w} \leftarrow \arg\max_V P_{i+1}$;
  $L \leftarrow L \cup \{\tilde{w}\}$;
**end**
**Return** $L$

---

# 4 Experimental Setup

This section elaborates on the experimental setup we adopt to evaluate our model and baselines on personalized code-mixed generation.

## 4.1 Datasets

To the best of our knowledge, no existing longitudinal dataset is available for Hindi-English code-mixed. A longitudinal dataset is required to study the temporal evolution of a language. Although some datasets in the literature consist of Hindi-English code-mixed texts collected from various online social networks, none of them contain user-specific information, making them unsuitable for our study. To overcome this, we collected code-mixed texts from the two most popular mediums where Indians are engaged – Twitter and YouTube. From Twitter, we collected over 0.8 million in tweets starting from the year 2011 till date, from which we filtered only tweets originating from Mumbai and Delhi metropolitan regions, two cities with the largest Hindi population. We used Twitter API for academic research with full archival access[1]. Further, for

---

[1]https://api.twitter.com/2/tweets/search/all

relevance, we restricted ourselves to tweets related to 'Cricket', 'Bollywood', 'Politics', and 'Government'. Starting at 2014, Twitter automatically tags the language of a tweet. We selected tweets with only non-empty language tags. This gives us a total of $226,480$ tweets from $19,782$ users.

From YouTube, we chose two channels – *NishaMadhulika*[2] (a popular chef based out of India with more than 12.7 million followers), and *T-Series*[3] (a popular Hindi music record channel started in 1983 having more than 200 million followers). We selected 42 videos from the NishaMadhulika channel and 69 from the T-Series that were first posted in 2011. We scraped all comments corresponding to these videos, accounting for $144,822$ comments from $99,998$ users.

For both datasets, we use a pre-trained language model open-sourced with Huggingface[4], that was fine-tuned on Hindi-English parts-of-speech (PoS) and language identification (LID) tasks. Using this model, we label each token in each text with the corresponding language (Hindi or English) and their associated PoS. We tag a text as code-mixed only when the text contains at least one Hindi verb written in either Devanagari or Roman script. We select users who have at least three utterances in their entire timeline. Finally, we are left with $18,126$ tweets (from $2,241$ users) and $8,957$ YouTube comments (from $1,349$ users).

We remove all the HTML tags, URLs, emoticons, user mentions (starting with '@'), and hashtags (starting with '#'). For simplicity, we remove all numeric values from texts, as well. Finally, we convert all texts to lowercase. We highlight the key statistics of the datasets in Table 1. We use a 75-25 split for training and validation with stratified sampling. Therefore, we can ensure at least one training and validation sample for each user. For the meta distillation curriculum, we split the training dataset in a $2:1$ ratio to generate the quiz dataset. Although the generative models can generate text with any starting seed word (*e.g.,* [CLS] token), to guide the code-mixed generation, we use the seed word as the first word of each validation utterance for each user.

## 4.2 Evaluation Metrics

We adopt intrinsic and extrinsic evaluation metrics to evaluate our model in terms of semantic understanding of code-mixing language and the ability to personify code-mixing for different users.

For the intrinsic evaluation, we use **perplexity**, a metric that measures the predictive power of a language model, compared against ground truth. We calculate perplexity as $e^{loss}$, with *loss* being the cross-entropy reconstruction loss on the validation data. A lower perplexity score indicates better reconstructibility and ability to learn the semantics of a generative model.

| Dataset | #Texts | #Users | Mean text length | Mean CMI |
|---------|--------|--------|------------------|----------|
| Twitter | 18126  | 2241   | 21.77            | 0.41     |
| YouTube | 8957   | 1349   | 28.89            | 0.36     |

Table 1: Dataset statistics, with mean text CMI and the average text lengths, demonstrating the extent of code-mixing.

Unlike Gupta et al. (2020), we do not have any labelled gold data for evaluating our generative model. Therefore, traditional supervised evaluation metrics – BLEU (Papineni et al., 2002), Rouge (Lin, 2004) can not be used directly to evaluate the personification aspects of code-mixed generation models. Similarly, other extrinsic evaluation measures such as Multilingual index (M Index) (Barnett et al., 2000), Burstiness and Span Entropy (Guzmán et al., 2017) can not be used, as these metrics are predominantly used to evaluate the ability to capture corpus-level switching patterns of generative models. To overcome the limitations of the existing evaluation metrics, we propose four metrics for benchmarking generated code-mixed texts against the historical utterances by different users. We devise **CM BLEU** by calculating the BLEU score between the candidate and reference language sequences. For example - consider a candidate code-mixed text "*mujhe park janaa hai*" ("I want to go to the park") and a reference text "*mujhe rice aur curry khana hai*" ("I want to eat rice and curry"). Using the LID model, we can extract the corresponding language sequences {Hi, Hi, Hi, Hi} and {Hi, En, Hi, En, Hi, Hi, Hi, Hi, Hi, Hi, Hi } from the candidate and reference texts, respectively (here, Hi and En stand for Hindi and English, respectively). Therefore, considering only the unigram and

---

[2] https://www.youtube.com/c/nishamadhulika

[3] https://www.youtube.com/aashiqui2

[4] https://huggingface.co/sagorsarker/codeswitch-hineng-lid-lince

bigram overlaps between the candidate and the reference, we calculate the CM BLEU score[5] of 0.606. If we use a different reference text "I don't want your nautanki" (Translation - "I don't want your gimmick") with the corresponding language sequence {En, En, En, En, Hi}, the CM BLEU reduces to 0.218. The proposed metric could calculate the similarity between the switching patterns demonstrated in the candidate text and the historical references by calculating the overlap between the language sequences. Similarly, we compute **CM Rouge-1** and **CM Rouge-L** by computing Rouge-1 and Rouge-L scores between the candidate and reference language sequences.

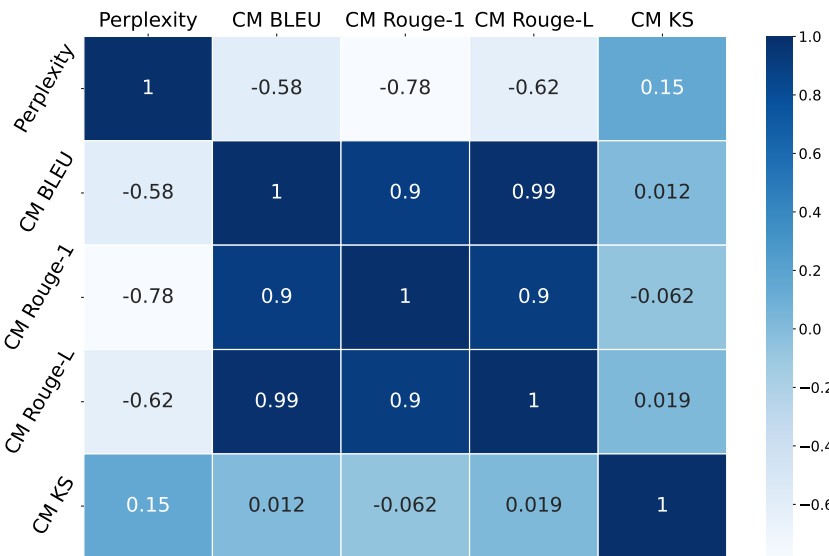

Figure 4: Pearson correlation between different evaluation measures on the validation dataset.

Additionally, we leverage the user-level historical CMI to evaluate the linguistic patterns of the generated texts. If a user historically prefers monolingualism over multilingualism, we want the generative model to learn the pattern and generate texts with a lower CMI value for the user. Towards this, we propose **CM KS**, a metric that computes the Kolmogorov-Smirnov distance between the generated and original CMI distribution of users. We highlight the relationships between these metrics by calculating the Pearson correlation between these measures, reported in Figure 4. Strong negative correlations between perplexity and CM BLEU, CM Rouge-1, and CM Rouge-L indicate that understanding semantics is essential to personify and replicate the switching patterns. Therefore, by learning semantics well, the generative models can learn the code-mixing patterns for different users and generate texts that imitate users' linguistic patterns. On the other hand, the correlations between CM KS and other metrics are weak, indicating that the linguistic preferences of users have no apparent linear relationships with switching patterns.

### 4.3 Baseline Methods

We consider several code-mixed generation models for comparative evaluation.

▷ **VACS** (Samanta et al., 2019) is a VAE-based encoder-decoder model, primarily developed for generating Hindi-English synthetic code-mixed texts.

▷ **GCM** (Rizvi et al., 2021) toolkit uses several linguistic theories and heuristics to generate code-mixed texts.

▷ **CM-XLM** (Gupta et al., 2020) is a generative model that utilizes pre-trained multilingual language model XLM to generate Hindi-English code-mixed texts from parallel corpora.

These code-mixed generation models consider monolingual reference data for generating code-mixed texts and generate code-mixed texts at a corpus level. Therefore, we compare these baselines only in intrinsic

---

[5]Can be calculated and validated using `https://www.nltk.org/api/nltk.translate.bleu_score.html`

| Model | Training Curriculum | Training Step | Optimizer | Learning rate | $\beta_1$ | $\beta_2$ |
|---|---|---|---|---|---|---|
| PARADOX | Vanilla | - | Adam | 4e-4 | 0.9 | 0.98 |
| PARADOX | Distillation | - | Adam | 4e-4 | 0.9 | 0.98 |
| PARADOX:Student | | Student evaluation | SGD | 4e-4 | - | - |
| PARADOX:Teacher | Meta Distillation | Teacher alignment | SGD | 4e-5 | - | - |
| PARADOX:Student | | Student relearning | Adam | 4e-4 | 0.9 | 0.98 |

Table 2: Optimizers for different models used for training.

```
[INST]
<<SYS>>
You are a helpful assistant that generate a Hindi-English text based on a user's previous message and a seed
word. Make sure that you understand the user's linguistic pattern from the previous message and generate a
text that starts with the seed word.
<</SYS>>

User Previous Message: {user_message}
Seed Word: {seed_word}
[/INST]
```

Figure 5: Prompt used with the Llama 2 model for Hindi-English code-mixed text generation.

evaluation. We also utilize several self-attention-based encoder-decoder and pre-trained language models to evaluate the personification aspects of generative models.

▷ **Transformer** (Vaswani et al., 2017) is an encoder-decoder architecture utilizing self-attention mechanism that has shown superior performances in generation tasks like machine translation.

▷ **MuRIL** (Khanuja et al., 2021) is an encoder-based language model pre-trained on 17 Indian languages with a masked language modeling objective.

▷ **BLOOMZ** (Muennighoff et al., 2022) is a family of large language model based on multilingual BLOOM (Workshop et al., 2023) that was fine-tuned with multitask prompting. We use the 3B parameter BLOOMZ model as our baseline.

▷ **Llama 2** (Touvron et al., 2023) is a family of auto-regressive large language models trained with reinforcement learning with human feedback (RLHF). We adopt the 13B parameter instruction-tuned model as one of our baselines.

▷ **GPT-4** (Achiam et al., 2023) is a large multimodal model, accepting image and textual data and generating text outputs through auto-regressive generation.

These pre-trained language models are only utilized in the extrinsic evaluation.

### 4.4 Training Details

For all the models across all the experiments, we use a maximum text length of 40. PARADOX consists of six encoder and decoder layers, with hidden sizes of 768 in all the layers. For multi-headed FAME and masked multi-headed FAME blocks, we use a total of eight heads with *Dropout* probabilities set as 0.1. The total number of parameters is 296M. We use six encoder and six decoder layers in the Transformer model, with eight heads in each multi-headed attention block. For training PARADOX, We set the persona encoding variational weight $\lambda = 0.5$. All the models are trained for 50 epochs with an early stopping condition on validation loss with the patience of 10. We set $batch\_size = 4$ in all experiments during training and validation. We fine-tune the MuRIL and BLOOMZ models on auto-regressive language modeling tasks for 10 epochs with learning rates $3e-5$ and $3e-6$, respectively. The Llama-2 baseline is used in zero-shot and 1-shot settings with the prompt shown in Figure 5.

We report all the different optimizers used for `PARADOX` in Table 2. We use the pre-trained MuRIL model as a teacher in the distillation and meta distillation. We use one Tesla P100 and one Tesla V100 GPU to run all our experiments. For `PARADOX`, each training and validation iteration takes $\sim 0.18$ and $\sim 0.12$ seconds, respectively. Strubell et al. (2019) proposed estimation of power usage and carbon emission behind running deep learning experiments. Following those guidelines, we estimate a total power usage of 23.56 kWh and an equivalent $CO_2$ emission of 22.46 pounds.

## 5 Comparative Analysis

In this section, we report the performances of `PARADOX` and the non-persona-based code-mixed generation models in terms of the intrinsic and extrinsic evaluation measures. We report the intrinsic evaluation results in Table 3. `PARADOX` achieves 43% better perplexity on the Twitter dataset than the vanilla Transformer. On the YouTube dataset, the margin is even higher (45%). A lower validation perplexity shows `PARADOX`'s strong ability to understand code-mixing semantics and generate texts of different linguistic variations. `PARADOX` achieves 18% better perplexity on the Twitter dataset than the best non-transformer baseline VACS. On the YouTube dataset, the margin is even higher (47%).

Table 4 highlights the extrinsic measures across all the generative models. On the Twitter dataset, `PARADOX` achieves 1.43 points better CM BLEU than the Transformer model. On the YouTube dataset, however, `PARADOX` without a contextual persona performs the best and outperforms the Transformer model with a margin of 1.31. In terms of the Rouge measures, `PARADOX` performs consistently bet-

| Model | Perplexity ↓ | |
|---|---|---|
| | **Twitter** | **YouTube** |
| GCM* | 4331.85 | 4323.15 |
| CM-XLM* | 5413.22 | 1603.59 |
| VACS | 361.05 | 552.35 |
| Transformer | 680.07 | 473.84 |
| (+) Distillation | 445.41 | 375.24 |
| (+) Meta Distillation | 797.84 | 506.82 |
| `PARADOX` | **297.43** | **292.44** |
| (+) Distillation | 377.64 | 294.87 |
| (+) Meta Distillation | 567.90 | 414.42 |
| (-) Contextual Persona | 320.79 | 295.26 |
| (-) Alignment | 337.07 | 363.77 |
| (-) FAME | 839.52 | 382.00 |

Table 3: Intrinsic evaluation of the competing models based on perplexity (↓: lower value indicates better performance). For models highlighted with *, perplexity is calculated with word-level generation.

ter than the non-persona counterpart with an average margin of 0.8 points. In terms of distance-based measures, `PARADOX` performs significantly better than the Transformer model on both datasets. Overall, `PARADOX` achieves 4% lower CM KS distance than Transformer. Lower KS distance indicates the importance of utilizing user persona in generating user-specific code-mixed texts.

It is interesting to notice the positive impact of distillation on extrinsic metrics. `PARADOX` meta-distilled with MuRIL achieves 0.88 points better CM BLEU score, on average. Similar improvements are observed with other extrinsic measures. Even Transformer meta-distilled with MuRIL achieves better CM BLEU, CM Rouge-1 and CM Rouge-L scores than the non-distilled ablation. Among the pre-trained language models, fine-tuned Llama 2 and GPT-4 are most competitive. Interestingly, even with a single example in the prompt (1-shot), CM BLEU increases by 9.8 points for Llama 2. Similar performance improvements are also observed with other extrinsic metrics. However, both Transformer and `PARADOX` perform significantly better than the pre-trained language models in terms of personalized code-mixed text generation. On average, `PARADOX` achieves 4.2 points better CM BLEU, 11 points better CM Rouge-1 and 9.6 points better CM Rouge-L than Llama 2. PARADOX achieves 13% better CM BLEU than the fine-tuned Llama model. Similar performance improvements are observed with CM Rouge-1 and CM Rouge-L metrics. In terms of the CM KS metric, the fine-tuned Llama model performs better than PARADOX. This highlights that the fine-tuned Llama model can preserve the population-level code-mixing patterns but fails to capture the personification aspect of it. PARADOX achieves 6.8% better CM Rouge-1 and 6.9% better CM Rouge-L than the GPT-4 model. Even with CM KS, our model outperforms GPT-4 with a wide margin of 18%.

Our ablation study shows the effectiveness of fused attention, contextual persona module, and the alignment module in `PARADOX`. Adding FAME improves validation perplexity by 57%. Similarly, the contextual user

| Model | CM BLEU ↑ | | CM Rouge-1 ↑ | | CM Rouge-L ↑ | | CM KS ↓ | |
|---|---|---|---|---|---|---|---|---|
| | Twitter | YouTube | Twitter | YouTube | Twitter | YouTube | Twitter | YouTube |
| MuRIL | 9.92 | 9.85 | 26.93 | 21.10 | 23.65 | 19.63 | 0.42 | **0.23** |
| BLOOMZ | 14.20 | 23.87 | 49.22 | 56.61 | 45.93 | 55.09 | 0.40 | 0.30 |
| Llama 2 (zero-shot) | 19.97 | 7.25 | 48.91 | 30.86 | 43.74 | 28.72 | 0.56 | 0.43 |
| Llama 2 (1-shot) | 26.69 | 20.03 | 55.17 | 46.08 | 49.57 | 43.17 | 0.50 | 0.39 |
| Llama 2 (fine-tuned) | 21.97 | 26.09 | 55.89 | 58.24 | 51.07 | 55.17 | **0.30** | **0.14** |
| GPT-4 (zero-shot) | **30.94** | 30.33 | 57.46 | 57.88 | 50.89 | 53.69 | 0.42 | 0.39 |
| Transformer | 22.21 | 29.36 | 58.69 | 61.02 | 51.10 | 57.27 | 0.42 | 0.37 |
| (+) Distillation | 22.52 | 29.68 | 59.35 | 61.30 | 51.91 | 57.45 | 0.38 | 0.39 |
| (+) Meta Distillation | 20.94 | 30.62 | 56.40 | 61.73 | 49.79 | 58.14 | 0.37 | 0.42 |
| PARADOX | 23.64 | 29.02 | 59.71 | 61.62 | 52.24 | 57.83 | 0.36 | 0.34 |
| (+) Distillation | 21.63 | 30.64 | 58.54 | 62.10 | 51.26 | 58.44 | 0.34 | 0.41 |
| (+) Meta Distillation | 24.38 | 30.04 | **60.79** | 61.90 | **53.18** | 58.52 | 0.32 | 0.48 |
| (-) Contextual Persona | 24.06 | **30.67** | 60.03 | **62.44** | 52.52 | **58.71** | 0.35 | 0.34 |
| (-) Alignment | 23.56 | **30.67** | 59.84 | 62.12 | 52.22 | 58.43 | 0.32 | 0.37 |
| (-) FAME | 20.37 | 29.36 | 55.78 | 60.80 | 49.16 | 57.37 | 0.36 | 0.37 |

Table 4: Extrinsic evaluation of pre-trained language models, Transformer and PARADOX in terms of preserving user-level switching patterns (↑ (*resp.* ↓): higher (*resp.* lower) value indicates better performance).

persona and alignment modules improve validation perplexity by 29% and 40%, respectively, justifying their contributions to modelling low-resource language. Therefore, having only the persona module is not sufficient for understanding the language of personalized code-mixing. However, we observe that having a contextual persona could hurt the ability of the model to preserve the switching patterns for different users. This could be attributed to users demonstrating minimal variability in their linguistic pattern, the trend we highlighted in Figure 1. Therefore, a random exploration with the contextual persona module could change the switching patterns observed for different users. In this case, reusing the static persona to generate texts could be semantically and linguistically more meaningful.

### 5.1 Human Evaluation

We perform a human evaluation study to evaluate the code-mixed texts generated by PARADOX and the vanilla Transformer. We randomly sample 24 examples from each of these models and ask 30 human evaluators[6] to rate these examples based on *Semantic coherence* and *Linguistic quality.* Semantic coherence measures the meaningfulness of the code-mixed texts, whereas, with linguistic quality, we measure their structural validity. Both the scores ranged between 1-5, 1 being the lowest, and 5 being the highest.

| Model | Semantic Coherence ↑ | Linguistic Quality ↑ |
|---|---|---|
| Transformer | 2.34 | 2.32 |
| PARADOX | **3.08** | **3.00** |

Table 5: Human evaluation of the models.

Table 5 presents the average semantic coherence and linguistic quality scores, along with Fleiss's Kappa (Fleiss, 1971) scores among the annotators. We observe that PARADOX displays a better semantic coherence (32% better), as well as better linguistic quality (29% better) than the Transformer model. We observe fair agreement (Kappa 0.13 for semantic coherence and Kappa 0.14 for linguistic quality) among the annotators for both models.

## 6 Analysis of Code-Mixed Generation

We further study the quality of code-mixed generation and compare them against other baselines. We analyze the distribution of length and CMI of texts generated by different generative models and report in Figure 6. Trained on the Twitter dataset, PARADOX with distillation generates texts with a median length of 20, 54% higher than the other generative baselines, and 17% higher than the actual. A similar trend can also be observed in the YouTube dataset. Similarly, the median value of CMI on texts generated by PARADOX is 25

---

[6]Evaluators are proficient with Hindi-English code-mixed language and their age ranges in $21 - 35$.

and 17, respectively, for the Twitter and YouTube datasets, which are significantly lower than the median CMI achieved by other baselines (30 and 28, respectively). Texts generated by `PARADOX` are even more monolingual than the actual dataset.

Figure 7(a) shows the distribution of top Hindi verbs and nouns from the Twitter dataset. Being more inclined towards monolingual, `PARADOX` assigns more probability to these Hindi words, irrespective of the parts of speech. Figure 7(b) shows the distribution of top Hindi verbs and nouns from the Twitter dataset for different ablations of `PARADOX`. Interestingly, `PARADOX` with the alignment module, can replicate the word distribution observed in the real dataset. On the other hand, without the alignment module, the generative model could hallucinate and unrealistically use common phrases in incorrect contexts. This highlights the effectiveness of the alignment module in recalibrating output tokens and generating semantically meaningful texts. Although the dimension of the alignment matrix is $|V| \times |V|$, with $|V|$ being the decoder vocabulary size, the learnable parameters are of order $O(d^2)$, where $d$ is the hidden dimension. The total number of additional parameters introduced by the alignment module is 0.025% of the entire network, which is insignificant compared to other modules.

We further analyze a few sample texts generated by the generative models. We report examples of texts generated by `PARADOX` and the Transformer, along with the average semantic coherence and linguistic quality scores annotated by the annotators in Table 6. Transformer usually picks top nouns in the corpus and generates texts around them without considering the syntax of the code-mixed language, resulting in incoherent texts in many cases. On the other hand, `PARADOX` preserves the grammar of code-mixed texts with a more human-like switching pattern. It shows that `PARADOX` maintains the grammar of the base language (Hindi in this case), attributing to a more coherent and reliable generation. The examples highlight the key differences between the texts generated by persona-based `PARADOX` and non-persona-based Transformer models regarding text quality. Table 7 further shows the personalized generation by our model. With the same prompt (*e.g.,* '*salman*' in the first example), `PARADOX` can understand different personas of users and can generate texts suited for different users. The high similarity between the historical average of the user CMIs and the generated CMIs indicates the model's ability to understand the linguistic preferences of users.

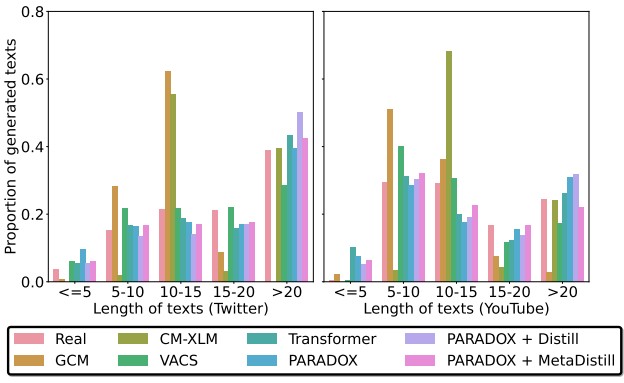

(a) Distribution of text length.

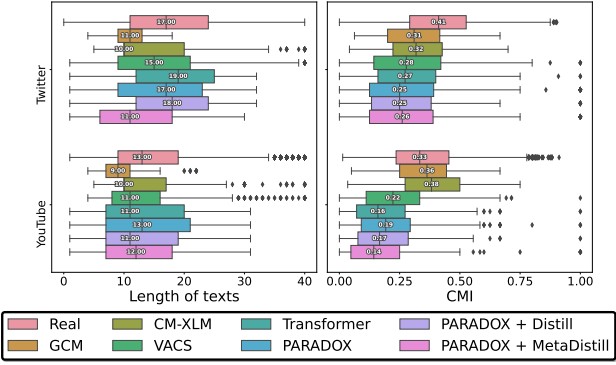

(b) Distribution of text length and CMI.

Figure 6: Comparison of texts generated by different generative models.

Table 8 highlights code-mixed texts generated by the Llama 2 model with zero-shot and 1-shot. `PARADOX` exhibits better capability in mimicking the code-mixing linguistic traits than Llama. For user ID 2226, who has had more monolingual usage in the past (average CMI 0.04), the text generated by Llama is more code-mixed than monolingual. Similarly, for user ID 3, the Llama model reverses the linguistic preference of the user while generating the code-mixed texts. Albeit demonstrating superior performance across various natural language understanding and reasoning tasks with zero and few-shot in-context learning, pre-trained

| Model | Generated Text | Semantic Coherence | Linguistic Quality |
|---|---|---|---|
| PARADOX | **CM:** are bhai apne! 
 **Eng:** Hey my brother! | 3.70 | 3.60 |
| | **CM:** bhai bahut thik ho gaya 
 **Eng:** brother is very well | 4.27 | 4.27 |
| Transformer | **CM:** rockstar walo se samjha media........ 
 **Eng:** Media understood by rockstars | 3.07 | 3.57 |
| | **CM:** boy likha hai thanks bhai 
 **Eng:** Boy has written thanks brother | 3.57 | 3.53 |

Table 6: Examples of code-mixed texts generated by PARADOX and Transformer with human annotated average Semantic coherence and linguistic quality scores.

large language models such as Llama fail to understand the linguistic complexities of informal languages such as code-mixed Hinglish. It is imperative to notice that the Llama model not only fails to capture the historical linguistic preferences of users but also fails to impersonate the semantic structure of code-mixed texts. On the other hand, PARADOX demonstrates better code-mixed language understanding capabilities, captures the linguistic preferences of the user from their historical utterances and preserves the information for future generations. This highlights the effectiveness of personification in code-mixed text generation and the importance of building more robust language understanding models for understanding the linguistic nitty-gritty of low-resource languages.

# 7 Conclusion

This paper described a personalized code-mixed generation model for generating human-alike code-mixed texts. We highlighted the need for a personalized generation under the pretext of code-mixing. Toward this, we devised a novel persona-aware encoder-decoder model coupled with a novel alignment module for generating more realistic and coherent Hindi-English code-mixed texts, the first attempt toward personalized code-mixed generation. Empirical analyses would benefit the research community in developing robust and reliable language models for low-resource languages. Although our empirical study has shown the effectiveness of persona-attributed text generation, currently PARADOX captures only contextual persona, ignoring other explicit factors. Not only does this restrict our model in cold-start generation (text generation for new users without any history), but it also fails to consider the co-association among users. In conversational settings, particularly, this can be deemed essential. Another limitation of PARADOX is the inability to determine the

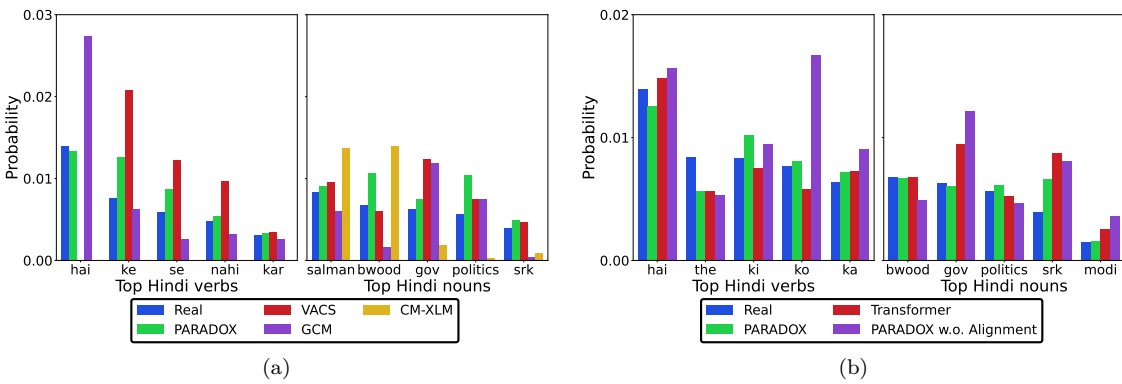

Figure 7: Distribution of top Hindi verbs and nouns for Tweets. w.o. alignment is the ablation of PARADOX without the alignment module.

| User ID | Generated Text | Historical Avg. CMI | Generated CMI |
|---|---|---|---|
| 3 | **CM:** salman ji please aapse milna hai
**Eng:** Salman, I want to meet you, please | 0.28 | 0.33 |
| 264 | **CM:** salman ka fan ho na.
**Eng:** You are Salman's fan, right? | 0.50 | 0.40 |
| 762 | **CM:** ye politics kr raha kya kr rahi h or truth show
**Eng:** Is this politics, or truth show | 0.15 | 0.30 |
| 2226 | **CM:** ye fb ho jaye bhaijaan
**Eng:** This has became fb, brother | 0.04 | 0.00 |

Table 7: Example of different prompted (prompts highlighted with blue) generation for different users. Different CMI indicates the difference in prompted generation based on the user persona.

| User ID | Model | Generated Text | Historical Avg. CMI | Generated CMI |
|---|---|---|---|---|
| 3 | `PARADOX` | **CM:** salman ji please aapse milna hai
**Eng:** Salman, I want to meet you, please | 0.28 | 0.33 |
| | Llama zero-shot | **CM:** Salman ne kaha tha flash me jeetega apka siddhant
**Eng:** Salman had said flushme jeetega apna principle | 0.28 | 0.22 |
| | Llama 1-shot | **CM:** Salman bhai ne kaha, Flush mein jitega apna side, main toh bus se nikal raha hoon!
**Eng:** Salman had said flushme jeetega apna side, I am just leaving in a bus | 0.28 | 0.11 |
| 2226 | `PARADOX` | **CM:** ye fb ho jaye bhaijaan
**Eng:** This has became fb, brother | 0.04 | 0.00 |
| | Llama zero-shot | **CM:** ye superb jabardast bahut khub, bhadaai ho
**Eng:** This is superb, amazing, and congratulations | 0.04 | 0.37 |
| | Llama 1-shot | **CM:** ye superb jabardast bahut khub, bhadaai ho
**Eng:** This is superb, amazing, and congratulations | 0.04 | 0.37 |

Table 8: Example of texts generated by `PARADOX` and Llama 2 models.

temporal evolution of a user's persona driven by external factors. `PARADOX` captures the user persona and its evolution solely from contextual information. At the same time, a user's linguistic preferences can also be driven by other external socio-demographic and economic factors varying over time, which our model currently undermines.

**Broader Impact Statement**

Our work highlights the need for personalized generation models for conversational languages like code-mixing. We release our curated dataset to encourage research on personalized code-mixed text generation. Persona-aware code-mixed generation models can aid in building data-driven solutions in low-resource languages and can be expanded to broader demographics. Although we do not anticipate any immediate negative impact of our work, over-personalization can lead to targeted spamming and misusing user persona negatively. We ask the researchers to be aware of the potential misuse and use the shared artefacts judiciously to prevent unwarranted events.

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
