# OpenReview forum: "Persona-aware Generative Model for Code-mixed Language"
_TMLR — Rejected by TMLR_

### Review · Reviewer_iZ7H · 2024-02-27

**Summary Of Contributions:**

This paper proposed a persona dialogue generation model, which can generate code-mixed dialogue responses considering the persona preference. The model is an encoder-decoder architecture, of which the encoder consists of a Transformer encoder and a contextual persona encoder, and the decoder consists of a Transformer decoder and an alignment module. The model is evaluated on Hindi-English dataset curated from Twitter and Youtube.

**Audience:**

Yes

**Claims And Evidence:**

No

**Requested Changes:**

1. My main concerns are mentioned in the above mentioned weakness.

2. Additional minor issues:

a. The starting paragraph of Section 3, "implicitly encode a user's persona based on her historical utterances". Why it's her? The work only models the woman persona?

b. In 3.1.1, Emb_u is modeled by a static embedding layer. What does static mean? This layer will be fixed?

c. The generation is initialized by a seed word. How to select a seed word? I guess this seed word will greatly influence the generation performance.

**Strengths And Weaknesses:**

**Strengths**:

The model can generate persona code-mixed dialogue response, which is prevalent on some social medias. I think this task is interesting.

**Weakness**:

1. Some of the architecture designs are not reasonable:

a. For the contextual persona encoder, why the author maps the persona embedding into a latent space instead of directly add it to the encoder hidden representations? Does the author want to improve the diversity of the generated texts by sampling different latent representations? However, according to Algorithm 3, it seems the generation is done only once.

b. For the token alignment, it seems like a self-attention with weights equal to embedding similarities, and a followed-up residue network. Does this violate the autoregressive rule as j-th word in the output sequence actually sees the similarity to all tokens in the sequence?

2. Some of the experimental settings are not reasonable:

a. The method is only evaluated on one dataset, curated Hindi-English dataset.

b. The automatic evaluation metrics are simple word overlap evaluation. Why doesn't the author consider embedding similarity metrics? As the analyse shown in Figure 4, all word overlap metrics have negative correlation with PPL, which validates those used metrics are not the best ones.

c. How does the author control the grammatical correctness and fluency of the curated dataset? As some examples shown in 4.2, it seems using simple word replacement can achieve the same results.

d. As mentioned in 4.2, the authors extract the corresponding language sequence. Isn't it more reasonable to use the token-level language tags to directly control the generated tokens?

---

> ### Author Response · Authors · 2024-03-07
> **Responses to Reviewer iZ7H Part-I**
>
> We thank you for the valuable feedback. We address the raised concerns. We have also incorporated our responses into the manuscript. All changes in the manuscript are highlighted in blue.
>
> **W1.a**
> In Section 3.1.2, we hypothesize that each user has a static (global) persona and a contextual (local) persona. The motivation behind projecting the persona embedding to a latent space is to capture the contextual perturbations in the user persona. For example, a user can have a generally positive outlook (global) on a particular concept; however, given a particular situation, the extent of the positive outlook (local) might change.
>
> **Changes to manuscript:** Section 3.1.2.
>
> **W1.b**
> We are obliged to clarify the confusion that arose. As explained in Section 3.1.4 and Section 6, we compute the dot-product $Q \cdot K^{T}$ over the space of decoder vocabulary *i.e.,* the attention matrix $\mathcal{A}$ lies in $\mathbb{R}^{|V| \times |V|}$. $V$ is the decoder vocabulary. This differs from auto-regressive attention, which computes the dot product over the temporal space *i.e.,* $\mathcal{A} \in \mathbb{R}^{L \times L}$, where $L$ is the length of the sequence. Unlike vanilla self-attention, the motivation behind the alignment module is not to calibrate the temporal sequences but to recalibrate the tokens, given their contextual similarity with other tokens.
>
> For sequence decoding, we still use masked self-attention (similar to the original Transformers), allowing us to generate texts auto-regressively.
>
> **W2.a**
> A persona-based code-mixed generation process requires high-quality code-mixed texts with the associated speaker/author information. As highlighted in Section 1, to our knowledge, there is no existing dataset on "user-level" code-mixed texts. Ours is the first large-scale dataset containing user-specific code-mixed texts obtained from social platforms.
>
> **W2b.**
> We are obliged to clarify the confusion that arose. We have adopted an intrinsic evaluation metric, perplexity and four extrinsic metrics, CM BLEU, CM Rouge-1, CM Rouge-L and CM KS. Perplexity calculates the likelihood of a generated text, whereas the extrinsic metrics evaluate the personification (linguistic comparison with the historical utterance of the same user) of generated code-mixed texts. Unlike Samanta et al., 2019 and Gupta et al., 2020, we don't have any ground truth validation dataset for evaluation. Therefore, we cannot adopt a similarity-based metric for evaluation.
>
> We have elaborated on the motivations behind the evaluation metrics in Section 4.2. Perplexity is calculated as negative log-likelihood. Therefore, a lower perplexity score is always desired. On the other hand, as highlighted in Table 4, the higher values of CM BLEU, CM Rouge-1 and CM Rouge-L indicate better preservation of users' linguistic demonstrations. A lower CM KS score indicates higher distributional similarities between the generated CMI distribution and the historical CMI distribution. Therefore, a negative correlation is expected between perplexity-CM BLEU, perplexity-CM Rouge-1 and perplexity-CM Rouge-L.
>
> **W2.c**
> Examples highlighted in Section 4.2 are hypothetical texts used to explain the designed evaluation metrics. We request the reviewer to go through Tables 6-8 for actual examples. Also, it is important to note that code-mixing is a conversational language often used on social platforms and conversational channels. Therefore, these real-life settings do not always preserve grammatical correctness and fluency.
>
> **W2.d**
> As the language tags are typically unavailable during decoding, one needs to extract the tags using some other means, which could impact the latency in inference. Additionally, instead of explicitly passing the language tags, our design enables the generative model to capture the linguistic patterns by looking at the user's historical utterances. Therefore, we claim no benefit of using these tags for controlled generation.
>
> **References**
>
> [Samanta et al., 2019] Samanta, Bidisha, Sharmila Reddy, Hussain Jagirdar, Niloy Ganguly, and Soumen Chakrabarti. "A deep generative model for code-switched text." arXiv preprint arXiv:1906.08972 (2019).
>
> [Gupta et al., 2020] Gupta, Deepak, Asif Ekbal, and Pushpak Bhattacharyya. "A semi-supervised approach to generate the code-mixed text using pre-trained encoder and transfer learning." In Findings of the ACL: EMNLP 2020.

---

> ### Author Response · Authors · 2024-03-16
> **Responses to Reviewer iZ7H Part-II**
>
> **C2.a**
> We apologize for the confusion that arose. We refer to the collective genders as 'her' as a common English practice. Our study is gender-neutral, and we do not use any gender-specific information in the curated dataset or the code-mixed generation model.
>
> **C2.b**
> Static embedding refers to an encoding technique where the embedding vector for each word does not change in different contexts. This is similar to any commonly used word or token embedding technique.
>
> **C2.c**
> A seed word is a token to initiate the generation process. Almost all of the generative models use seed words (*e.g.,* [CLS]) to initiate the auto-regressive decoding. For model evaluation, we chose the seed word from the validation dataset (see Section 4.1). For each user in the validation dataset, we use the first word in their utterance as the seed word. Therefore, we can evaluate the linguistic pattern of the generated text against the ground truth text in the validation dataset.
>
> **Changes to manuscript:** Section 4.1.

---

### Review · Reviewer_JrxA · 2024-03-10

**Summary Of Contributions:**

This paper proposes a personalized code-mixed generation model for generating human-alike code-mixed texts. The key motivation is to incorporate user information while generating code-mixed texts. To this end, this paper makes a pioneering effort in utilizing user persona by leveraging a Transformer-based encoder-decoder architecture to learn the semantics of code-mixed generation and utilizing a novel persona encoder to encode a user persona from their behavioural preferences. Instead of projecting the user’s persona onto a static space, the proposed approach projects it onto a probabilistic latent space and captures the contextual persona based on their historical persona. Additionally, the approach uses an alignment module to re-align decoder outputs to generate coherent texts. In contrast to existing studies, the proposed approach does not require parallel corpora or reference data for code-mixed text generation.

**Audience:**

Yes

**Broader Impact Concerns:**

This paper has the Broader Impact Statement section. The contents generally make sense.

**Claims And Evidence:**

Yes

**Requested Changes:**

Please see Weaknesses above.

**Strengths And Weaknesses:**

Strengths:

1. The paper is clearly written and the proposed technique is sound.

2. This work makes a solid contribution by generating personalized code-mixed texts by leveraging users’ historical utterances. The research direction is timely and the motivation is reasonable.

3. The experiments are comprehensive and the results well indicate the effectiveness of the proposed approach.

Weaknesses:

1. For the main results in Table 4, the comparison is mainly based on zero-shot and 1-shot ICL of Llama. The comparison could be unfair as the proposed approach is a fine-tuned one. Besides, the results has shown inferior than the ICL Llama. It is interesting to see if the proposed approach is effective in those large generative models.

2. The alignment module does yield marginal performance gains. It is better to have a discussion about the results.

---

> ### Author Response · Authors · 2024-03-16
> **Responses to Reviewer JrxA**
>
> We thank you for the valuable feedback. We address the raised concerns. We have also incorporated our responses into the manuscript. All changes in the manuscript are highlighted in blue.
>
> **W1.** We thank the reviewer for suggesting additional experiments to highlight the effectiveness of our proposed method. We have evaluated our model against the fine-tuned Llama 2-13B model. The results are mentioned below.
>
> |      Model       | CM BLEU                                || CM Rouge-1                                   || CM Rouge-L || CM KS ||
> | :---------------- | :------: |  :------: |  :------: |  :------: |  :------: |  :------: |  :------: | ----: |
> |             | Twitter                                | YouTube                                   | Twitter                                   | YouTube                                | Twitter | YouTube | Twitter | YouTube |
> | Llama 2 (fine-tuned) | 21.97                                          | 26.09                                              | 55.89                                              | 58.24                                           | 51.07            | 55.17            | **0.30**             | **0.14**             |
> | PARADOX              | **24.06**                                           | **30.67**                                              | **60.79**                                              | **62.44**                                           | **53.18**            | **58.71**            | 0.32             | 0.34             |
>
> PARADOX achieves 13% better CM BLEU than the fine-tuned Llama 2-13B model. Similar performance improvements are observed with CM Rouge-1 and CM Rouge-L metrics. In terms of the CM KS metric, Llama 2-13B performs better than PARADOX. This highlights that the fine-tuned Llama model can preserve the population-level code-mixing patterns but fails to capture the personification aspect of it.
>
> **Changes to manuscript:** Table 4 and Section 5.
>
> **W2.** Although the results highlighted in Table 4 in the main text suggest a relatively marginal contribution of the alignment module, Table 3 indicates that the alignment module contributes an average 18% improvement in perplexity. Therefore, the alignment module is crucial in generating semantically meaningful and coherent code-mixed texts. Additionally, Figure 7b and our discussion in Section 6 suggest that proper alignment is necessary for the generative model to prevent assigning unrealistic weightage to different words, preventing incoherent generation.

---

### Review · Reviewer_6dsJ · 2024-03-12

**Summary Of Contributions:**

Code-mixing data generation is often user-dependent, but previous studies often ignore the impact of the user. This paper explores explicitly modelling user persona based on user’s historical data to improve code-mixing data generation.

Contributions are three folds:
1) Dataset: the authors collected two datasets from Twitter and Youtube with user information for Hindi-English code-mixing generation.
2) Modelling: the authors propose a Transformer-based encoder-decoder model with user pearsona contextualization module and target alignment module, which achieves better performance than baselines.
3) Evaluation: the authors propose several surface-based new evaluation metrics and adopt human evaluation to examine the generation quality.

**Audience:**

Yes

**Broader Impact Concerns:**

This paper collected user data from twitter and youtube. The research may be conducted without user consent but I'm not sure whether this is necessary.

**Claims And Evidence:**

No

**Requested Changes:**

1. In section 3.1.3, the authors state “... input sequence concatenated with positional encoding …”. Transformer often adopts addition rather than concatenation. Is this a typo? or why concatenation is preferred here?
2. In section 3.1.1, the authors mentioned the use of “batch normalization”. Did you use batch normalization or layer normalization?
3. Since user’s historical comments are leveraged to model pearsona, why also add user ID information explicitly? Would the performance become significantly worse after removing user ID? If so, does that mean the model itself can’t infer user’s features from data directly?
4. As user ID is adopted, how did you perform inference for new users?
5. In section 3.2.2, the call the normal training of student “student evaluation”? what’s the “quiz data”? and what’s the teacher’s prompt?
6. Based on Table 4, it seems the contextual pearsona helps little on generation quality. What are the additional benefits by using latent probabilistic modelling? What does the latent variable capture?
7. I’m not sure whether using user data from Twitter and Youtube is permissive for research. but since the authors plan to release them, what data licence you will use and what constraints when using the data?
8. How did you get the seed dataset for the evaluation? How does it affect the performance? How many examples are in the test set?
9. The experiment only uses 1 user utterance for peasona modelling. What if  using more utterances?
10. For large language models, please consider adding more powerful comparisons such as gpt4. Also, please compare with llama2 with more number of few-shots.
11. The proposed evaluation metric only considers the code-mixing behaviour without checking the data content, and the generation is very open-ended. It doesn’t make much sense to me.
12. While human evaluation is performed, the evaluation only focuses on quality without examining the key focus of this study: peasona. Did the model really perform generation reflecting the users features?

The authors need to reconsider these concerns carefully.

**Strengths And Weaknesses:**

Strengths:
* The idea of personalising code-mixing generation is interesting
* The constructed dataset, proposed modelling and evaluation could facilitate the future research in this direction

Weaknesses:
* Model description is not always clear
* Concerns regarding user consent and licence remain for the benchmark
* Ablation study should be improved
* Evaluation has problem

---

> ### Author Response · Authors · 2024-03-16
> **Responses to Reviewer 6dsJ Part-I**
>
> We thank you for the valuable feedback. We address the raised concerns. We have also incorporated our responses into the manuscript. All changes in the manuscript are highlighted in blue.
>
> **RC1.** We thank the reviewer for bringing this to our notice. This is an unintended mistake; we use the addition of the token embeddings and the positional encoding.
>
> **Changes to manuscript:** Section 3.1.3.
>
> **RC2.** We apologize for this typo. We use layer normalization, which is also used in the Transformer model. Thank you very much for pointing this out.
>
> **Changes to manuscript:**  Section 3.1.1.
>
> **RC3.**  The user ID information is passed as an identifier to the model, as the training batch might contain utterances from different users. Therefore, without an identifier, the model cannot understand whether the same user or a different user makes an utterance. The model learns a static persona embedding for each user ID and derives the contextual persona based on the current utterance. Without the user ID, our method can not generate personalized texts, as it cannot determine the historical persona.
>
> **RC4.** Although for cold-start generation, we can use our proposed method along with a placeholder static persona embedding, the generated text may not capture the true linguistic patterns of the user. This is a limitation of our work highlighted in Section 7. In future, we plan to leverage the external persona factors to demonstrate users for personalized text generation.
>
> **RC5.** Section 3.2.2 elaborates on the meta-knowledge distillation (MetaDistil) curriculum. In MetaDistil, in step 1, we perform a "student evaluation", equivalent to vanilla student training without distillation. In step 2, the teacher model is fine-tuned w.r.t. a distillation loss calculated between the teacher and the frozen student model on a smaller sample of the training data, which we refer to as "quiz data". Finally, in step 3, the student model is further fine-tuned on the task-specific and distillation loss, calculated against the frozen teacher model. We refer to the "teacher's supervision" as the "teacher prompt" to the student.
>
> **Changes to manuscript:**  Section 3.2.2 and Section 4.1.
>
> **RC6.** In Section 3.1.2, we hypothesize that each user has a static (global) persona and a contextual (local) persona. The motivation behind projecting the persona embedding to a latent space is to capture the contextual perturbations in the user persona. For example, a user can have a generally positive outlook (global) on a particular concept; however, given a specific situation, the extent of the positive outlook (local) might change.
>
> Table 3 highlights that contextual persona is crucial for enhancing the semantic quality of the generated text. The contextual persona module contributes up to 7% increase in the perplexity of generated texts.
>
> **Changes to manuscript:**  Section 3.1.2.
>
> **RC7.** We would like to clarify that in our collected dataset, we only use the user utterances and the anonymized user ID. Additionally, the dataset is cleaned, and all personal addresses are removed to maintain user privacy. The additional user information is neither collected nor used in the personalized code-mixed generation. We intend to open-source the dataset under the CC BY 4.0 licensing agreement.
>
> **RC8.** For model evaluation, we chose the seed word from the validation dataset (see Section 4.1). For each user in the validation dataset, we use the first word in their utterance as the seed word. Therefore, we can evaluate the linguistic pattern of the generated text against the ground-truth text in the validation dataset.
>
> **Changes to manuscript:**  Section 4.1.
>
> **RC9.** We consider only the users with more than two utterances in the dataset, out of which the last utterance is used for evaluation, and the rest is for training. Therefore, in the training dataset, all users have more than one utterance for training the personalized code-mixed text generation model.
>
> **Changes to manuscript:**  Section 4.1.

---

> ### Author Response · Authors · 2024-03-16
> **Responses to Reviewer 6dsJ Part-II**
>
> **RC10.** We thank the reviewer for suggesting additional experiments to highlight the effectiveness of our proposed method. We have evaluated our model against the fine-tuned Llama 2-13B model. Additionally, we have compared our model with GPT-4. The following results are obtained from the fine-tuned Llama and GPT-4 models.
>
> |      Model       | CM BLEU                                || CM Rouge-1                                   || CM Rouge-L || CM KS ||
> | :---------------- | :------: |  :------: |  :------: |  :------: |  :------: |  :------: |  :------: | ----: |
> |             | Twitter                                | YouTube                                   | Twitter                                   | YouTube                                | Twitter | YouTube | Twitter | YouTube |
> | Llama 2 (fine-tuned) | 21.97                                          | 26.09                                              | 55.89                                              | 58.24                                           | 51.07            | 55.17            | **0.30**             | **0.14**             |
> GPT-4 (zero-shot) | **30.94** | 30.33 | 57.46 | 57.88 | 50.89 | 53.69 | 0.42 | 0.39
> | PARADOX              | 24.06                                           | **30.67**                                              | **60.79**                                              | **62.44**                                           | **53.18**            | **58.71**            | 0.32             | 0.34             |
>
> PARADOX achieves 13% better CM BLEU than the fine-tuned Llama model. Similar performance improvements are observed with CM Rouge-1 and CM Rouge-L metrics. In terms of the CM KS metric, the fine-tuned Llama model performs better than PARADOX. This highlights that the fine-tuned Llama model can preserve the population-level code-mixing patterns but fails to capture the personification aspect of it. PARADOX achieves 6.8% better CM Rouge-1 and 6.9% better CM Rouge-L than the GPT-4 model. Even with CM KS, our model outperforms GPT-4 with a wide margin of 18%.
>
> **Changes to manuscript:**  Table 4 and Section 5.
>
> **RC11.** As elaborated in Section 4.2, the intrinsic evaluation metric, perplexity, measures the semantic coherence of the generated texts. The extrinsic and proposed evaluation metrics measure the personification aspects of the generated texts.
>
> **RC12.** As evaluating the personification of texts entails subjectivity, we refrain from adopting human evaluation for this assessment. Instead, we employ human annotators to assess the generated texts' semantic coherence and linguistic quality. The proposed extrinsic measures -- CM BLEU, CM Rouge-1, CM Rouge-L and CM KS, reported in Table 4, assess the generative models' personification capabilities. As we do not have any user-specific features for evaluation, we only compute the similarity between historical and generated linguistic patterns with these extrinsic metrics.

---

### Author Response · Authors · 2024-03-23
**Requesting reviewers for checking our rebuttals**

Dear Reviewer 6dsJ,  JrxA and iZ7H,

We have addressed all the concerns you raised. Kindly review our rebuttals and let us know if you have any further questions.

---

### Decision · Action_Editor_yKb4 · 2024-04-24

**Recommendation:** Reject

**Comment:**

Two of the three reviewers found that the paper did not meet the acceptance criteria at TMLR and did not recommend acceptance. We discussed further internally, and the reviewers identified precise points they found problematic in the latest version of the manuscript. I outline these criticisms below. These suggest more than minor modifications, and I'd be happy to consider a revised version of the work for publication at TMLR.

- User ID. The need to condition on a user ID (instead of, say, the previous utterances of a user) limits the method to "warm-start" users. It's possible that it would be enough to recognize this limitation early in the paper.
- Contextual persona module. The reviewers found the parametrization of this module, and, in particular, the latent space modelling, to require additional justification given the claims made in the paper (notably that it "capture[s] the contextual perturbations in the user persona"). The reviewers recognized that the module was shown to provide generation improvements (Table 3) but wondered how critical the proposed approach was in obtaining these results. For example, one reviewer suggests running an ablation where a simple neural network replaces the latent modelling part to start exploring this question.
- CM evaluation metrics. The claim that the proposed CM-based evaluation metrics evaluate the personalization is dissonant with what the methods seem to capture. In particular, the proposed evaluation metrics seem to evaluate a restricted version of personalization that doesn't consider which tokens are code-mixed. Perhaps clarifying the definition of personalization would help align these ideas.


More minor:
Terminology. The paper seems to introduce new terms (including "student evaluation," "teacher alignment," and "quiz data") that may not be standard. I am passing this here because I think it might be helpful to the authors, but it is minor, and I don't think it's impossible that these terms have been used in the literature.
- Language tags. This comment in the author's response, "We claim no benefit of using these tags for controlled generation," made reviewers curious about the motivation for using them in the first place.
- Regarding the possibility of releasing the dataset. My understanding is that it might depend on the license of the original providers (YouTube and Twitter?).
- Evaluation dataset(s). There are passages in the manuscript that read as if you had collected a single dataset (e.g., "we collect a large-scale longitudinal dataset"). I'd suggest clarifying that you have collected and are studying the proposed method using two datasets.
   - It would also be interesting to limit the current scope of the method to the Hindi-English domain and, ideally, discuss how the proposed model might generalize to code-mixed data/users using different language pairs.

**Audience:**

This paper discusses current NLP¨methods and would interest the TMLR audience.

**Claims And Evidence:**

Overall, some elements of the evaluation do not seem to support the claims made in the paper.

**Resubmission Of Major Revision:**

The authors may consider submitting a major revision at a later time.